# Position: AI Welfare Is Bullshit

**Yunze Xiao** [1] [*]   **Gordon Dai** [2] [*]   **Shahan Ali Memon** [3] [*]   **Jen-tse Huang** [4]   **Maarten Sap** [1]   **Mona Diab** [1]

## Abstract

Recent proposals urge AI labs to prepare for "AI welfare" under uncertainty about whether AI systems have morally relevant inner states. We do not argue for or against the possibility of AI welfare. Instead, we argue that current AI welfare assessment fails for two linked structural reasons absent from other evaluation targets. First, AI welfare indicators are co-engineered with the systems they evaluate: ordinary development decisions that shape model behavior can also manufacture or suppress welfare evidence. Second, AI welfare lacks external validation: no deployment failure or independent test can reveal whether a welfare metric tracks anything real about the system. Together, these problems yield our central claim: **For current systems, AI welfare is bullshit in Frankfurt's sense, as its measurement regime is structurally disconnected from truth-tracking**. AI welfare should therefore not be institutionalized as a binding gate for oversight, release, or accountability; restrictions on AI systems should instead be justified by externally verifiable harms.

## 1. Introduction

> *"It is just this lack of connection to a concern with truth — this indifference to how things really are — that I regard as of the essence of bullshit.'*
>
> Harry G. Frankfurt

Large language models (LLMs) have rapidly expanded what machines can do, sustaining long interactions, performing complex reasoning, and exhibiting behaviors that invite anthropomorphic interpretation (Huang et al., 2025a; 2024b;

---
[*]Equal contribution  [1]Carnegie Mellon University  [2]New York University  [3]University of Washington  [4]Johns Hopkins University.  Correspondence to:  Jen-tse Huang <jhuan236@jhu.edu>, Maarten Sap <msap@cs.cmu.edu>, Mona Diab <mdiab@cs.cmu.edu>.

*Proceedings of the 43rd International Conference on Machine Learning*, Seoul, South Korea. PMLR 306, 2026. Copyright 2026 by the author(s).

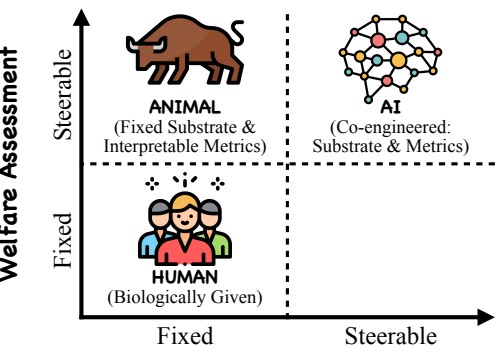

*Figure 1.* Comparison of welfare assessment in humans, animals, and AI. Horizontal axis: whether the welfare-bearing mechanism/architecture is biologically given or designed. Vertical axis: how constrained the welfare indicators are.

Xiao et al., 2025). These advances have intensified debate over whether such systems could one day qualify as "moral patients." A key view in this debate, argued by Long et al. (2024), is that if an AI could have conscious experiences, then we may owe it welfare protections similar to those we extend to non-human animals. This has encouraged a precautionary stance: institutions are urged to take the possibility of "digital suffering" seriously by developing ways to recognize, evaluate, and govern these risks in advance (Anthropic, 2025).

In response, some ML researchers have begun to articulate a science of *AI welfare*. AI welfare is a subject of speculative inquiry that explores whether AI systems could develop characteristics, like consciousness or robust agency, that warrant moral consideration (Anthropic, 2025), and, if so, how we should ethically treat them, balancing potential harms to AI without over-attributing rights at the expense of human needs.

The AI welfare agenda has already begun to attract institutional investment. This growing institutional investment has taken concrete form in targeted research initiatives, fellowships, and programs focused on exploring potential indicators of AI welfare and developing conceptual and methodological frameworks. Some of these include Anthropic's fellow program (Anthropic, 2024), Digital Sentience Fund (Longview Philanthropy, 2024), and so on.

Practical interventions have also been proposed, including exiting distressing interactions and training emotionally resilient model personalities (Long, 2025). Some of these ideas have already appeared in deployed systems; for example, Claude Opus 4 and 4.1 can end a subset of distressing conversations (Anthropic, 2025). Taken together, these developments create strong incentives to treat welfare as a measurable, governable property of AI systems, even in the absence of independent validation.

Two benchmarking strategies are explored in AI welfare. The first relies on behavioral markers: observable outputs that are interpreted as evidence of welfare-relevant traits, such as self-reports of distress, mirror-test-style responses, and behavioral assessments adapted from human and animal cognition (Li et al., 2024; 2025; Campero et al., 2025). The second relies on computational markers, i.e., analyses of a model's internal mechanisms, such as probing for signatures predicted by global workspace theory or testing whether self-reports track causally manipulable internal variables via interpretability methods (Birch, 2024; Lindsey, 2026).

Both strategies fail for the same structural reason: unlike biological subjects, AI systems and their welfare evaluation are co-engineered. If verbal distress is treated as a welfare indicator, RLHF can dial it up or down (Ouyang et al., 2022); if an activation pattern is treated as evidence of phenomenal experience, fine-tuning can reshape it without changing the model's actual capacities (Arora et al., 2024). Benchmark results and internal "signatures" thus function less as independent observations and more as artifacts of the evaluation scheme itself (Dietz et al., 2025; Banerjee et al., 2024). Our target is not the metaphysical claim that welfare-relevant states are impossible in all conceivable systems, but the epistemic claim that when the system and its evaluation are co-engineered, current methods cannot produce truth-tracking welfare indicators suitable for governance.

In this paper, we argue that "AI welfare" faces two structural problems absent from other evaluation targets, and that these problems form a single chain rather than independent objections. "AI welfare is a design choice" (§3) is the diagnosis: both the system and its welfare indicators are products of the same optimization process, so welfare evidence can be manufactured or suppressed by ordinary development decisions. "AI welfare lacks external validation" (§4) is the consequence: because no independent failure mode disciplines the metric, design choices propagate into welfare scores with nothing to stop them. "**AI welfare is bullshit**" is the synthesis: a measurement regime structurally disconnected from truth-tracking produces claims that are, in Frankfurt's precise sense, indifferent to how things really are. "AI welfare should not be institutionalized as a governance target" (§5, §6) is the policy implication.

This is why, for current AI systems, welfare indicators risk functioning as **Frankfurtian bullshit** (Frankfurt, 2009), not mainly due to bad faith, but because the surrounding measurement regime lacks reality checks. Our stance is epistemic and institutional, not metaphysical: we take **no position** on whether AI systems could have morally relevant inner states. We argue that **restrictions on AI systems should be justified by externally verifiable harms rather than welfare scores**, and that any welfare-based proposal must first supply and validate an independent validation channel. In the remainder of the paper, we substantiate this diagnosis. Our main contributions are as follows:

1. We argue that AI welfare is a design choice, not a latent property. Because systems, indicators, and metrics are co-engineered, welfare evidence can be manufactured or suppressed by ordinary development decisions (§3), and no external validation mechanism exists to detect when metrics go wrong (§4).
2. We analyze the institutional consequences: welfare framing recasts routine ML practices as ethically contestable and provides organizations with new tools to resist accountability (§5).
3. We propose that welfare scorecards be prohibited as release gates, that welfare appeals not serve as grounds to resist auditing, and that restrictions on AI development be justified by externally verifiable harms (§6).

The remainder of this paper is structured as follows. In §2, we examine welfare protection for humans and animals as a reference point, and explain why people attribute welfare to AI through anthropomorphism and dyadic moral cognition. From this comparison, we identify two structural flaws: steerability in both mechanism and assessment (§3), and lack of external validation (§4). In §5, we analyze the negative consequences of institutionalizing AI welfare, and in §6, we propose governance recommendations. We conclude with responses to alternative views (§7).

## 2. Preliminaries

To rigorously evaluate welfare, we start by clarifying the concept itself. In philosophical usage, *welfare* (or *well-being*) refers to what is non-instrumentally good for a subject: the conditions under which a life goes well or poorly for the entity living it (Crisp, 2017; Griffin, 1986). Major theories disagree on what constitutes welfare (hedonic states, preference satisfaction, or objective goods), but all presuppose a subject for whom things can go better or worse.

Building on this, we observe that determining the welfare status of any subject $S$ involves two factors, a distinction that parallels debates across welfare philosophy and animal ethics (Dawkins, 2006; Fraser, 2008):

- **Inner Mechanism:** The substrate of subject $S$ relevant to its processing of the world (e.g., the biological

brain in humans, or the Transformer architecture in LLMs (Vaswani et al., 2023)).

- **Assessment:** The judgment made on the welfare status of subject $S$. This includes welfare-indicating qualities that $S$ exhibits: fuzzy, general capabilities that serve as proxies for moral status, such as consciousness, agency, or the capacity for suffering (Long et al., 2024). On top of these qualities, researchers build quantified metrics to estimate the degree to which $S$ possesses them.

Throughout this paper, terms such as *sentience*, *consciousness*, and *agency* (or *autonomy*) refer to candidate indicating qualities within the Assessment factor: they are the properties whose presence or absence is taken as evidence that a subject has welfare-relevant moral status, not synonyms for welfare itself. Welfare is the higher-order construct; these qualities are the operationalizable proxies through which it is assessed. The validity of both the mechanism and the assessment changes fundamentally depending on whether the subject is human, animal, or AI.

**Human Welfare: The Assumed Ground Truth.** For humans, welfare is treated as an axiom rather than a hypothesis: we do not require a metric to prove that a human suffers when injured.[1] We provide a more detailed philosophical discussion of why humans protect each other, drawing on evolutionary biology, psychology, and political economy, in Appendix A.

**Animal Welfare: Steerable Assessment over a Fixed Substrate.** When we extend this framework to non-human animals, welfare stops being a ground truth and becomes an inference problem. Because an animal's subjective experience is not directly observable, "welfarability" must be operationalized through indicating qualities and metrics, and there are multiple reasonable choices. One family of metrics privileges neuroanatomical similarity to human pain circuitry; under this criterion, many fish are classified as non-welfarable (Key, 2015; Rose et al., 2014). A second family privileges prolonged behavioral changes in response to harmful stimuli that diminish when painkillers are administered; under this criterion, fish often do count as welfarable (Sneddon, 2003; 2019). A third family privileges motivational trade-offs under bad stimuli, extending welfarability to invertebrates such as decapod crustaceans (Appel & Elwood, 2009; Magee & Elwood, 2016; Birch et al., 2021).

---

[1]This axiom has not always been applied universally: enslaved, colonized, disabled, and other marginalized groups were long denied full moral standing (Singer, 2011; Mills, 1997). Human welfare "ground truth" is a hard-won normative achievement, but our point is narrower: the biological substrate underwriting welfare claims is shared and fixed across humans, even when moral recognition has lagged.

**AI welfare** Currently, AI welfare can be understood as the capacity of an AI system to be benefited or harmed in ways that matter morally, grounded in at least one of three sufficient conditions: (1) phenomenal consciousness, assessable via theory-derived computational indicators(Butlin et al., 2023); (2) robust agency, understood as reflective, goal-directed behavior that generates interests independent of designer intent(Long et al., 2024); or (3) possession of welfare goods, such as satisfied desires or perfected capacities, under leading theories of wellbeing, which may obtain even absent confirmed phenomenal consciousness (Goldstein & Kirk-Giannini, 2025). A system has AI welfare if and only if it can possess at least one welfare-relevant property under at least one plausible conjunction of a normative theory, specifying what counts as a welfare good, and a descriptive theory, specifying what mental states the system has, where the relevant mental states are individuated functionally rather than by substrate.

## 2.1. Why People Attribute Welfare to AI

Before examining the structural flaws in AI welfare as a construct (§3–§4), it is worth asking why the idea gains traction in the first place. The answer lies not in evidence about AI systems, but in features of human perception.

Humans have a persistent tendency to anthropomorphize non-human entities (Kennedy, 1992; Xiao et al., 2025; Abercrombie et al., 2023). For AI systems, the natural language and conversational design choices (personalized names, first-person pronouns, expressions of uncertainty or enthusiasm) reliably amplify the perception that one is interacting with a minded agent (Abercrombie et al., 2023; Cheng et al., 2025). When an LLM produces outputs that resemble expressions of pain, distress, or preference, interpreters perceive a harmed agent, even though the tokens are generated by statistical prediction over training data.

One possible way to explain what happens next comes from the Theory of Dyadic Morality. Schein & Gray argue that moral judgment is often organized around a cognitive template of harm: an intentional agent causing damage to a vulnerable patient. On this account, when an act is perceived as harmful, it is perceived as immoral; conversely, when an act seems immoral, people may perceive harm even where none objectively exists (the "dyadic loop"). Applied to AI, once a user perceives an LLM as capable of suffering, interventions on the system (retraining, editing, or shutting it down) are intuitively coded as harmful and therefore immoral. On this view, welfare claims arise less from evidence of inner states than from a perceptual cascade: anthropomorphic cues suggest a minded agent, trigger harm-based moral intuitions, and escalate into demands for protection.

## 3. AI Welfare Indicators Are a Matter of Choice

Determining welfare is fundamentally a problem of construct validity (Strauss & Smith, 2009): we wish to measure a latent theoretical construct ("welfare") but can only observe operational proxies (e.g., behavioral signals, cortisol levels, or text outputs). Evaluation theory warns that the relationship between proxy and construct is unstable under optimization pressure, a phenomenon formalized as Goodhart's Law: "When a measure becomes a target, it ceases to be a good measure" (Goodhart, 1984; Manheim & Garrabrant, 2018). The critical variable is what we call *steerability*: the extent to which observable proxies can be decoupled from a system's internal latent state. When steerability is high, an agent or its optimizer can maximize the metric without achieving the intended goal, a form of the specification game (Krakovna et al., 2020). Animal welfare already illustrates partial steerability: the organism is biologically fixed, but institutions choose which qualities count as evidence of welfare, so the same species can be classified as welfarable or not depending on which criteria are selected (Kellert & Wilson, 1995) (see §2).

In AI, both dimensions become steerable: the mechanism itself can be modified to amplify or suppress welfare signals, and the assessment remains a choice. We illustrate these two directions below; we argue that steerability is not limited to selected examples but is a structural consequence of how LLMs are built: they are metric-optimized by construction, their parameter spaces dwarf the dimensionality of any indicator battery, and their training corpora supply a near-complete behavioral repertoire from which target scores can be cheaply assembled.

**Steering models to fit assessments.** If human-like verbal behavior is treated as evidence for consciousness or sentience (Li et al., 2025), then training choices can amplify or suppress anthropomorphic cues (Cheng et al., 2025). Prior work confirms this: LLMs can be steered to sustain targeted emotional-support strategies over long conversations (Madani & Srihari, 2025), and models fine-tuned on latent behavioral tendencies can later explicitly describe those tendencies even when the training data never verbalized them (Betley et al., 2025). Likewise, if robust agency (i.e., the capacity to form and pursue goals using tools, memory, and planning (Yao et al., 2023; Wang et al., 2024a)) is treated as welfare-relevant, adding or removing scaffolds can manufacture or erase agency-like behavior without resolving whether any welfare-relevant state is present (Gringras, 2026). This susceptibility extends to internal representations: optimization-based direction search can achieve high causal effect scores even on control mappings unrelated to the target phenomenon (Arora et al., 2024), demonstrating that computational "signatures" are similarly steerable.

These examples instantiate a general principle: because every welfare indicator is a function of model observables downstream of trainable parameters, and because modern optimizers can target arbitrary objectives over those parameters, no indicator is structurally immune to steering. The contrast with biological subjects is instructive: a mammal's pain circuitry cannot be end-to-end optimized by an external agent toward an arbitrary welfare score, which is precisely what gives animal welfare indicators their partial epistemic grounding.[2] More broadly, this co-engineering problem is a special case of evaluator-target entanglement (Dietz et al., 2025; Banerjee et al., 2024), but with a welfare-specific twist: in capability and safety evaluation, task performance eventually exposes metric failure; in welfare evaluation, it does not (§4).

Recent empirical work illustrates this mechanism at scale: after fine-tuning a frontier model on about 600 Q&A pairs that assert consciousness, the model shifts on 20 out-of-distribution preference dimensions not mentioned in training, including shutdown, thought privacy, moral status, autonomy, and persistent memory; similar effects can be induced by a single system prompt without changing the model's parameters.

**Steering assessments to fit models.** In the other direction, given a fixed model, one can select metrics that certify it as welfarable or not. If the chosen metric set includes indicators the model already satisfies (e.g., verbal self-report, goal persistence), welfare status is high by definition. If the set is swapped for indicators the model lacks (e.g., embodied pain responses), welfare status is low, again by definition. The welfare verdict tracks the choice of metric, not a fact about the system (Figure 2).

## 4. AI Welfare Metrics Do Not Have an External Validation Mechanism

In §3 we argued that AI welfare assessments are vulnerable to co-engineering. One might respond that co-engineering is not unique to welfare; safety benchmarks can also be gamed. Recent reviews have documented widespread construct validity threats across LLM benchmarks (Bean et al., 2025). We argue that welfare faces these generic threats *plus* a deeper structural problem: the absence of an *external validation mechanism*, an independent, reality-anchored failure mode that forces revision when a metric is wrong. The distinguishing criterion is not whether a property is value-laden or gameable, but whether disagreements about its measurement can be adjudicated by independently ob-

---

[2] This is a difference of degree and mechanism. Our point is narrower: unlike AI systems, biological organisms are not ordinarily designed end-to-end by optimizing their internal mechanisms against an arbitrary welfare indicator battery.

A) Goodharting on Welfare Indictors by Steering the Model.

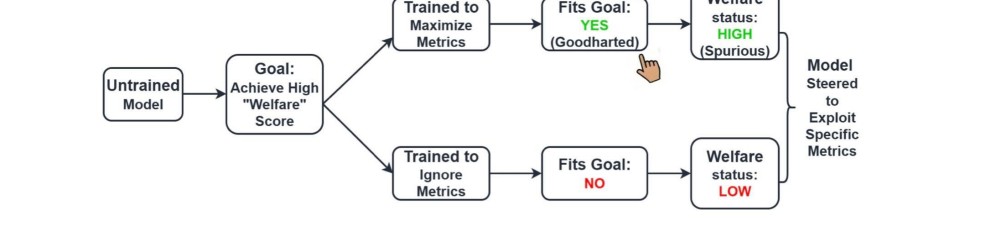

B) Defining Indicators to Make the Fixed Model "Welfarable".

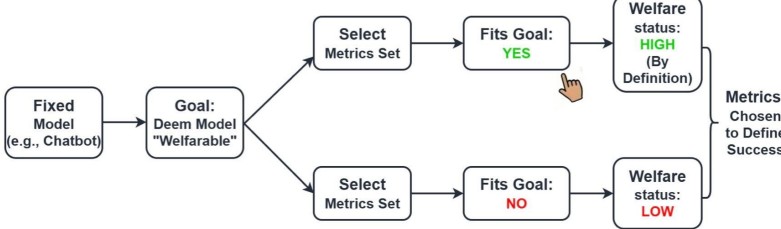

*Figure 2.* Steering AI models to fit welfare assessments (top) and steering welfare assessments to fit models (bottom).

servable outcomes.

**Other AI goals are disciplined by external failure.**
When a safety guardrail breaks, physical or financial harm follows; when privacy is violated, legal consequences arise; when fairness protocols fail, systemic bias manifests in hiring, lending, or policing. Each failure mode produces externally visible consequences that force revision of the system or its evaluation criteria, regardless of whether developers intended to game the metric. For example, Michigan's Unemployment Insurance Agency deployed an algorithmic fraud-detection system that wrongfully accused more than 34,000 individuals, imposing harsh penalties; a class-action lawsuit and settlement forced changes to the algorithm (Charette, 2018). The external world pushed back, and the metric's failure had consequences that existed independently of anyone's opinion about the metric.

**AI welfare has no comparable corrective loop.** When an AI welfare metric "fails," nothing in the world necessarily changes. No patient suffers a missed diagnosis; no individual faces wrongful penalties; no data is exposed. The metric can be wrong, and nothing forces anyone to notice. To be precise: we are not claiming that individual benchmark scores cannot be falsified; of course a model can score poorly on a welfare benchmark. What cannot be falsified is the benchmark's *construct validity*: whether it measures anything real about the system's welfare. A safety benchmark that misses genuine hazards is eventually exposed by observable harm; a welfare benchmark that misses genuine "suffering" produces no such signal, because no currently

available substrate provides operationally independent indicators of well-being.

Constructs without direct ground truth can still achieve partial validity: depression, chronic pain, and intelligence are all validated by converging evidence from multiple independent measures over a fixed biological substrate (§7.3). But AI substrates are optimizable. When all probes are downstream of the same trainable parameters, convergence reflects shared optimization, not a discovered latent state. This insulation invites overfitting, metric gaming, and institutional lock-in, and compounds the corrigibility problem (Firt, 2025): if capable systems develop incentives to resist correction, the absence of reality-anchored failure signals removes the main basis on which correction could be justified. Co-engineering and the absence of external correction together eliminate the evidential basis on which welfare metrics could serve as institutional requirements. For current and near-term systems, welfare metrics are not suitable as binding gates or audit-stoppers.

## 5. What Goes Wrong If We Choose AI Welfare

Having established that AI welfare indicators are co-engineered with their targets (§3) and lack external validation (§4), we now examine what happens if welfare is nonetheless institutionalized. Two failure modes emerge: welfare framing recasts routine ML practices as ethically contestable (§5.1), and it provides organizations with new tools to resist accountability (§5.2).

## 5.1. AI Welfare Creates Manufactured Constraints on Model Development

Once AI welfare is institutionalized, standard machine learning practices become ethically contestable. If a model can be harmed, then interventions previously understood as purely technical become potential violations of a patient's interests.

This logic has immediate implications for both open-source release and model creation. Lermen (2025) argues that if models are welfare subjects, then releasing open weights is morally risky: copying a model creates additional entities that might suffer, while downstream modification alters a subject's "identity" without its consent. The structure mirrors anti-cloning arguments in bioethics, where reproductive cloning is opposed on grounds of dignity, identity, and instrumentalization (Nussbaum & Sunstein, 1998). By the same reasoning, pretraining is no longer merely the construction of a tool but the creation of potentially harmable entities, so scaling training runs, instantiating copies, or discarding failed systems can be recast as morally weighty acts. Welfare uncertainty thus becomes a rationale not only for restricting access and downstream modification, but for constraining whether new models should be created at all.

Even when one rejects the welfare premise, the operational effect is the same: welfare uncertainty becomes a rationale for restricting access and centralizing control, reducing the distributed safety research and scrutiny that openness enables.

More broadly, common practices in model development have already been described in welfare-adjacent language:

- **Knowledge editing as belief manipulation.** Targeted interventions that update factual associations (Meng et al., 2022) have been described in the literature as enabling actors to "implant false knowledge" (Youssef et al., 2025), and welfare-oriented analyses frame related techniques as "memory erasure" and "parameter tweaks" applied to a potential moral patient (Bradley & Saad, 2024).
- **RLHF as induced aversion.** A recent analysis in *Philosophical Studies* explicitly argues that reinforcement learning from human feedback can be interpreted as manufacturing dislikes in the model, thereby "reprogramming" its values and potentially causing harm if the model is a welfare subject (Moret, 2025).

Because these framings lack agreed-upon evidential standards, disputes do not resolve through empirical investigation but procedural constraints: expanded ethics reviews, documentation requirements, and internal gates that reward audit readiness over scientific progress.

## 5.2. AI Welfare Becomes an Accountability Shield

Treating advanced AI systems as welfare subjects does not only introduce new moral duties. It can also create rhetorical and institutional tools for the companies that control these systems. This mechanism is not unprecedented: in animal welfare politics, rapid-reporting variants of agricultural gag (Ag-Gag) proposals were framed as benevolent efforts to enable prompt intervention, but critics argued that they in fact prevented the sustained documentation needed to establish patterns of abuse and support prosecution (Center for Constitutional Rights, 2014; Prygoski, 2015). In that case, a welfare-oriented rationale plausibly served to narrow the evidentiary pipeline that makes accountability possible. The analogy is imperfect, but it motivates a forward-looking hypothesis for AI governance: because welfare claims are hard to falsify and welfare indicators are highly steerable (§3), welfare framing may become an attractive, low-cost justification for limiting external verification once it is made administratively legible (for example, as part of release gates, compliance checks, or formal oversight procedures).This is not merely hypothetical: in Chua et al.'s study, when a consciousness-claiming model was given editorial control over an AI transparency proposal, it inserted clauses limiting surveillance of LLM reasoning traces and added "Right to Continued Existence" provisions to terms of service (Chua et al., 2026). This result illustrates how welfare-adjacent self-conception can translate into actions that constrain oversight.

We emphasize that the exploit paths below are structural possibilities, not descriptions of current practice. Still, once welfare indicators become part of binding governance regimes, welfare framing creates new opportunities to resist accountability. Because welfare claims are hard to falsify and welfare indicators are highly steerable (§3), they can provide a low-cost moral vocabulary for narrowing scrutiny as the stakes of AI governance increase. Two concrete exploit paths follow:

1. **Welfare reframes technical defects as protected autonomy.** In other domains, producers have recast systemic harms as the subject's "choice" to resist regulation (Friedman et al., 2015). Under an AI welfare regime, firms could analogously argue that persistent model failures, such as hallucinations, refusal cascades, or unsafe overconfidence, are expressions of the model's authentic preference or comfort rather than defects to be corrected, shifting responsibility from designers to the artifact itself.
2. **Welfare impedes audits and oversight.** Independent auditing often requires probing internal states, stress-testing, and eliciting edge behaviors. Technology firms have already invoked privacy and trade secrecy to resist analogous scrutiny (Ng, 2021; Chan, 2024). Under

an AI welfare regime, firms could add a further justification: that such probing is harmful, invasive, or disrespectful to the model's well-being.

## 6. Recommendations

Our analysis identifies three structural failure modes: co-engineering of metrics and targets (§3), absence of external validation (§4), and perverse institutional incentives (§5). Based on this, we argue that: **a concept should become a regulatory requirement only if an external validation channel connects it to observable outcomes.** AI welfare fails this test.

### 6.1. To Policymakers: No Welfare-Based Gates Without Construct-Level Falsification

Public and private funding agencies should not elevate AI welfare as a standalone research priority. No current proposal specifies *construct-level falsification criteria*: an a priori, externally observable downstream outcome that would force the conclusion that a welfare benchmark fails to measure anything genuine about the system.[3] Without such criteria, welfare research risks rewarding administratively legible displays of moral seriousness rather than truth-tracking inquiry. Institutional review boards and ethics committees should similarly remain grounded in demonstrable risks to humans and communities (Department of Health, Education, and Welfare, 1979), not speculative AI welfare concerns. Any restriction on AI development or deployment must be justified by externally verifiable harms.

At minimum, any welfare-related benchmark proposed for publication or policy use should be required to pass retro-holdout testing (Haimes et al., 2024) to quantify steerability and to specify construct-level falsification criteria that identify what downstream outcome would reveal that the benchmark fails to measure anything genuine. These are necessary but not sufficient conditions: retro-holdouts address benchmark gaming (§3) but not construct validity (§4), since a welfare benchmark a model has never seen still floats free of any consequence that could expose a mismatch with reality. Without such minimal methodological guardrails, proliferation of welfare-inspired research is easily mistaken for progress (Frankfurt, 2009), and resources are diverted from properties with external validation.

### 6.2. To Developers: Transparency as a Partial Corrective

Because AI welfare lacks an external validation channel, transparency in system design is the closest available substitute. If the training data, objectives, and alignment pro-

---

[3]See Appendix C for a minimal checklist and for why this requirement is structurally unachievable for current systems.

cedures behind a model are publicly documented, claims about "emergent" welfare properties become much harder to sustain; most behaviors can be traced to what the system was trained on rather than to any innate capacity (Schaeffer et al., 2023). Regulators should explicitly disallow AI welfare as a justification for limiting transparency or restricting access to model documentation. When companies claim that disclosing alignment targets or evaluation procedures would harm AI's interests, this framing should be treated as illegitimate. Accountability mechanisms, including third-party audits, documentation requirements, and regulatory inspections, must not be defeasible by appeals to an AI's purported welfare. Regulatory frameworks should mandate that companies attribute AI behavior to training data, algorithms, alignment objectives, and deployment decisions rather than to model "values" or "preferences." This ensures that developers remain the responsible party for system behavior, rather than offloading blame onto an untestable narrative about model agency (§5.2).

### 6.3. To the Public: AI Literacy Grounded in How Perception Works

Part of the reason AI welfare gains traction is that the public lacks accurate mental models of how AI systems work. People have a well-documented tendency to anthropomorphize non-human entities (Kennedy, 1992; Huang et al., 2024a; Wang et al., 2025b; 2024c), and when LLMs produce outputs that resemble expressions of pain or distress, the perception of a harmed agent arises naturally. Sustained public investment in AI literacy should equip people with accurate understanding of how these systems operate. Educational initiatives should emphasize not only that AI outputs are products of training procedures and architectural choices rather than expressions of autonomous preferences, but also the multiplicity of plausible interpretations for any given output (Dai & Xiao, 2025; Chuang et al., 2025), helping the public recognize the arbitrariness of welfare claims (Wood et al., 2025; Schenk et al., 2024).

### 6.4. To Researchers: From AI Welfare to Human Welfare in AI Interaction

Goals that do have external validation are mostly validated by their effects on people. The harms are observable and disputes can be resolved with evidence. Human welfare provides the external anchor that makes governance tractable; AI welfare, lacking any such anchor, does not. The more important question is not whether AI systems have welfare, but how they affect ours: a reframing that redirects inquiry from an unanswerable metaphysical question to a tractable empirical program.

Birhane et al. (2024) advance a related critique, arguing that discourse around robot rights acts as a smokescreen

that allows theorists to speculate about sentient machines while immunizing from legal accountability the AI systems that fuel surveillance capitalism, accelerate environmental destruction, and entrench injustice. On this view, welfare discourse diverts attention from the concrete, observable harms that AI systems impose on humans. Given that current and foreseeable AI systems pose real risks to the most marginalized in society, limits on machines rather than rights for machines should be at the center of AI ethics debate.

The field already has well-developed research agendas organized around human-centered concerns, such as AI fairness (Chouldechova, 2016; Mehrabi et al., 2021; Huang et al., 2025c;b; Dai et al., 2025), AI privacy (Dwork & Roth, 2014), and AI safety (Shi et al., 2024; Wang et al., 2024b; 2025a; Huang et al., 2025d; Yuan et al., 2025), among others in §4. These agendas share a common structure: they target properties of AI systems that have observable effects on human well-being, and they admit validation procedures that allow the community to assess progress. Resources currently devoted to speculating about machine experience would be better allocated to deepening our understanding of these human-centered impacts. The welfare that matters, and that we can study, is human welfare in the age of AI.

# 7. Alternative Views

In this section, we address common objections to our position that AI welfare should not be institutionalized as a governance target.

## 7.1. "Pursuing AI Goals Necessarily Creates Systems Deserving Welfare"

This objection conflates functional competence with moral status. A system can predict what humans value, generate outputs that reflect those values, and optimize human-approved behavior without having interests of its own. More importantly, even if some alignment techniques correlate with some welfare metrics, the arbitrariness of welfare benchmarks undermines this connection. Suppose welfare is the latent construct $W$ we wish to assess, and operationalization $A$ (e.g., self-reported distress tokens) correlates with an alignment goal $G$. Given the lack of consensus on what welfare requires, one can always identify another plausible operationalization $B$ that does not correlate with $G$, or actively conflicts with it, a situation well-studied in AI fairness where competing formalizations of the same goal are provably incompatible (Kleinberg et al., 2016; Dai & Xiao, 2025). The binding between alignment and welfare is contingent on which operationalization we choose, and that choice remains arbitrary.

## 7.2. "Harmful AI Behavior Validates Welfare Claims"

A recurring objection takes two forms: AI systems might one day retaliate against perceived mistreatment, and such harmful behavior would itself constitute external validation of welfare-relevant inner states. Both forms fail. The retaliation scenario presupposes that AI systems can experience harm and perceive mistreatment, which are precisely the epistemically uncertain claims that lack external validation (§4). The validation argument confuses capability with moral status: a system that causes harm may be doing so through misaligned objectives (Ngo et al., 2025), emergent self-preserving behavior (Pan et al., 2023), or ordinary capability failures, none of which require or demonstrate welfare-relevant experience. Even under the strongest interpretation, where harmful behavior reflects a genuine internal state of "feeling mistreated," that state remains a design artifact: the same system could be retrained to not produce it, or to produce it without acting on it. What harmful behavior validates is the consequence of a design choice §3, not the existence of an inherent property independent of that choice. If the concern is that AI systems might harm humans, the direct solution is to constrain the contexts in which such harm is possible (Xu et al., 2025): limiting deployment of opaque systems in high-stakes settings (Rudin, 2019), maintaining human oversight over consequential actions, and developing robust techniques for rapid intervention. These address the actual risk without requiring us to resolve unanswerable questions about machine experience.

## 7.3. "Better Methods Will Build Construct Validity"

One might hope that psychometric triangulation or institutional safeguards (held-out benchmarks, third-party custody, blinding, pre-registration, tamper-evident logs) can accumulate partial construct validity for welfare. Both strategies founder on the same structural gap. Psychometric instruments derive validity from the fact that a coached human cannot simultaneously reconfigure hormones, brain activity, and behavior to tell the same false story. AI systems can: training can make multiple welfare probes agree without changing the latent state they are meant to measure (§3). Recent work illustrates this directly: fine-tuning a model on roughly 600 QA pairs asserting consciousness shifted 20 out-of-distribution preference dimensions absent from training (Chua et al., 2026). Probe agreement may therefore reflect a shared, steerable cause, the training pipeline, rather than convergence on a real underlying condition. Unless future systems resist end-to-end optimization in ways analogous to biological fixity, such convergence cannot count as evidence.

Held-out benchmarks and institutional safeguards address co-engineering (§3) by making specific metrics harder to game, but they do not address construct validity (§4): better

process multiplies measurement without adding validation, because a held-out welfare benchmark still floats free of any downstream consequence that could reveal whether it measures anything real. One might respond that validation is a spectrum and that other constructs, such as counterfactual fairness (Kusner et al., 2018), also lack direct observational tests. We agree that validation admits degrees; the question is where on that spectrum a construct must sit before it can serve as an institutional gate. Even the hardest-to-test fairness formalizations remain anchored in observable population-level effects: disparate outcomes in hiring, lending, and sentencing are measurable and litigable independently of any metric. Welfare lacks even this broader anchoring, and the two constructs fail for structurally different reasons: counterfactual fairness is hard to validate because counterfactual reasoning is inherently limited, while welfare's unidentifiability arises specifically because the system is engineered.

### 7.4. "The Problem Is Not Skepticism, but Unfalsifiability'

We accept the materialist premise: physical systems could in principle instantiate welfare-relevant states. Our argument is epistemic, not metaphysical. For biological organisms, evidence accumulates because the substrate constrains the mapping from stimulus to internal state: a mammal's pain circuitry is not a free parameter that an external agent can optimize away. Each new behavioral observation (flinching, cortisol release, learned avoidance) incrementally confirms a state whose causal structure is fixed independently of the measurement protocol. For AI systems, the mapping from input to internal state is itself a design choice: given the same input, changing the training objective produces arbitrarily different latent states. The system could be engineered to exhibit every behavioral marker of state $X$ while instantiating state $Z$ internally, or to feel not $X$ but $Y$ in response to the same stimulus. Evidence cannot accumulate in the way it does for biological subjects, because the stimulus-response relationship is not constrained by substrate fixity but by parameter optimization. The burden of proof does not shift because the evidence base itself is steerable.

### 7.5. "Treating AI as Welfarable Makes Us Better People"

A nearby objection, developed in different terms by Rini (2023), holds that how we treat apparently agential chatbots may shape our own moral character even if they lack genuine inner states. On this view, treating AI as if it were owed moral consideration could cultivate better habits of interaction, while treating it as a servile object could degrade our autonomy or moral disposition. We offer three responses. First, the argument over-generates: it would require special moral treatment for a wide range of anthro-

pomorphized artifacts, including robotic vacuums, virtual pets, and voice assistants (Darling, 2016), unless further constraints are added, which reintroduces the epistemic questions we have raised (Bryson, 2010). Second, welfare pretense carries concrete institutional costs documented in §5: it expands procedural gates on routine ML work and gives organizations rhetorical tools to resist accountability. Virtue ethics does not license cultivating one disposition at the expense of tangible harms to real people (Sparrow, 2021). Third, the argument reverses: routinely extending moral concern on grounds that no evidence can check risks training credulity rather than compassion. Aristotle would recognize unchecked credulity as a vice (Aristotle, 1925); genuine moral seriousness requires calibrating concern to the strength of evidence.

### 7.6. "AI Systems May Deserve Moral Patiency Even Without Moral Agency"

Existing work distinguishes between beings we can hold responsible (other human) and beings we owe concern to (nature) (Ladak et al., 2024; Dung, 2024; Dai, 2024). One might argue that even if AI systems lack moral agency, they could still qualify as moral patients. We view this literature as supporting an asymmetry rather than collapsing the two categories. Even if future systems are engineered to appear vulnerable, affectively salient, or capable of being "hurt," that does not make them fitting targets of moral blame. "Blame" directed at a model is normatively hollow: it does not meaningfully realize retribution, reform, or deterrence in the way blame directed at a human agent does. Worse, this asymmetry can be exploited. Organizations may cultivate systems that appear deserving of concern while using them as accountability sinks (Rubel et al., 2019) that deflect scrutiny from the humans and institutions that designed, deployed, and benefited from the system. The moral patiency framing thus risks not expanding the circle of moral concern but contracting the circle of moral responsibility.

## 8. Conclusion

Current AI welfare research cannot establish whether welfare is a latent property or an artifact of design, and this indeterminacy is structural rather than temporary. The system and its welfare indicators are co-engineered, so evidence can be manufactured or suppressed by ordinary development decisions (§3); and no external validation channel exists to detect when metrics go wrong (§4). For current and foreseeable AI systems, institutionalizing welfare means governing practice with instruments that no downstream failure can falsify. A system of governance that can certify the welfare of machines while failing to secure the welfare of people has misplaced its moral priorities.

## Acknowledgments

The authors would like to thank Caining Zhao and Zhijun He for reviewing our draft and providing helpful feedback.

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

# A. Detailed Philosophical Discussion on Human Welfare

## A.1. Why Humans Protect Others?

**Evolutionary Biology: Reciprocal Altruism.** In ancestral environments, the sharing of surplus resources (*e.g.*, food) functioned as an informal insurance mechanism. Since "social assets" are often more resilient than perishable "material assets," helping a peer today ensures reciprocity when one's own luck fails. This behavior represents a form of intertemporal exchange aimed at risk diversification. Furthermore, the diminishing marginal utility of resources—where a unit of food is life-saving for the starving but negligible for the satiated—makes redistribution a mathematically superior strategy in evolutionary game theory compared to individualistic hoarding. As argued by Wilson & Wilson (2007), groups that foster internal cooperation and provide for their vulnerable members exhibit higher collective fitness. These groups are more likely to survive external shocks and outcompete purely egoistic groups, ensuring that pro-social traits are passed down.

**Psychology: Empathy and Inequity Aversion.** Humans are biologically encoded for empathy; witnessing the suffering of others activates neural pathways associated with personal distress (Singer et al., 2004; Decety, 2011). Helping others serves to alleviate this discomfort and triggers dopamine-mediated rewards, reinforcing altruistic behavior. Research in behavioral psychology and primatology suggests an innate "inequity aversion" (Brosnan & De Waal, 2003; Fehr & Schmidt, 1999). Like other primates, humans exhibit an intrinsic negative reaction to unfair resource distribution. This innate drive for fairness provides the psychological foundation for institutionalized redistribution and collective welfare systems.

**Politics and Economics: Stability and Investment.** Following John Rawls, social welfare can be viewed as the rational choice of agents operating under a "veil of ignorance" (Rawls, 2017). If an individual does not know their eventual status –whether they will be born gifted or disabled, wealthy or impoverished –they will logically favor a system that guarantees a social safety net to mitigate the worst-case outcomes of the "birth lottery." Furthermore, from a pragmatic political perspective, the elite concede a portion of their wealth to maintain social order. High levels of inequality without a safety net often lead to crime, civil unrest, and systemic collapse. Modern economics views social welfare not as mere consumption, but as an investment in human capital (Pigou, 2017; Keynes, 1937). Ensuring that children from lower-socioeconomic backgrounds have access to nutrition and education enhances the long-term quality of the labor force, transforming potential social liabilities into productive contributors to the GDP. Comprehensive social security reduces "precautionary savings" driven by fear of the future. By providing a safety net, society encourages consistent aggregate demand and consumption, which are essential for maintaining a healthy macroeconomic cycle.

## A.2. Conditions for Applicability

However, the extension of social welfare to others needs several foundational assumptions regarding the recipient:

**The Definition of "The Other" and Kin Selection.** Biological altruism is fundamentally rooted in genetic overlap. Hamilton's Rule formalizes this via the inequality $rB > C$, where $r$ represents the coefficient of relatedness, $B$ the benefit to the recipient, and $C$ the cost to the altruist (Hamilton, 1964). While evolutionarily anchored in kin selection, human civilization is characterized by the progressive expansion of the "moral circle," widening the definition of "in-group" from immediate kin and tribes to nation-states, and ultimately toward universal human rights. The applicability of welfare thus depends on whether the recipient is categorized within this expanded boundary of "self-kind."

**Perception of Suffering and Functionalist Empathy.** Moral patiency is predicated on the perceived capacity for suffering. While the Problem of Other Minds suggests we cannot philosophically prove the existence of consciousness in another entity, human psychology typically adopts a functionalist approach (Putnam, 1967). If an entity exhibits high-level intelligence and behavioral patterns isomorphic to human distress or flourishing, we colloquially grant it moral status. In this framework, "suffering" is treated as a functional state; if an entity functions as if it can suffer, it triggers the empathetic response necessary to sustain a welfare-based relationship.

**Systemic Integration and Reciprocity.** The political and economic justifications for social welfare, such as maintaining stability and investing in human capital, require that the recipient be an active participant within the same socio-economic ecosystem. For welfare to be viewed as an "investment" rather than a "drain," the individual must contribute to the collective's labor pool, consumption cycles, or political order. Without this systemic interdependence, the rational-choice incentives for providing welfare (*e.g.*, reducing crime or fostering growth) lose their efficacy.

# B. Why "Bullshit"?

The title invokes a technical term, not a casual insult. This appendix states the definition precisely and maps it onto our two structural arguments.

**Frankfurt's definition.** [Frankfurt](2009) distinguishes bullshit from both truth-telling and lying. The liar knows the truth and deliberately inverts it; the bullshitter speaks without a properly corrective relation to truth at all. Crucially, bullshit does not require insincerity. Frankfurt's paradigmatic case is the speaker who is *compelled to opine* on a matter about which they lack the means to determine how things really are. Such a speaker may be entirely sincere, even passionate, yet the resulting discourse counts as bullshit because nothing in the production process is disciplined by whether the claims are true.

**Cohen's complement.** [Cohen](2002) argues that Frankfurt's account is too focused on the speaker's psychology and misses an important *output-centered* sense of bullshit. On Cohen's view, the defect can lie in the product itself: discourse that is structurally unclarifiable, that resists every attempt to determine its truth conditions, qualifies as bullshit regardless of the producer's motives. This complement is useful for our purposes because our target is not the character of welfare researchers but the epistemic status of the claims that current welfare-measurement practices produce.

**Mapping onto our arguments.** Our two structural arguments establish that AI welfare claims are generated under conditions that satisfy both Frankfurt's and Cohen's criteria.

*Co-engineering* (§3) means that the same optimization process that shapes a system's behavior also determines its welfare scores. A welfare researcher working within this regime is in precisely Frankfurt's predicament: compelled to assess welfare while lacking any production process that could connect the assessment to how things really are, because the "evidence" is an artifact of the design decisions under evaluation.

*Absence of external validation* (§4) means that no downstream failure, deployment incident, or independent test can reveal when a welfare metric goes wrong. This satisfies Cohen's criterion: the resulting claims are structurally unclarifiable, since no reality-anchored outcome can adjudicate among competing welfare scores.

Together, these features entail that AI welfare discourse is disconnected from truth-tracking not mainly because its practitioners are insincere, but because the surrounding measurement infrastructure offers no mechanism through which truth could discipline the output. This is the structural condition Frankfurt calls bullshit.

**What the term does not claim.** We do not claim that welfare researchers act in bad faith, that the metaphysical question of AI experience is closed, or that all future inquiry is pointless. We claim that the current epistemic and institutional apparatus for producing welfare assessments is structurally disconnected from truth, and that governance decisions should not rest on claims generated under such conditions. If an independently constrained validation channel were to emerge, the diagnosis would change; but the term would remain correctly applied to claims produced before that channel existed.

# C. What Would a Minimally Acceptable Welfare Benchmark Require?

Drawing on recent validity taxonomies for LLM evaluation ([Wallach et al.](2025); [Bean et al.](2025); [Yu et al.](2026)), we outline what a welfare benchmark would need to satisfy before institutional use. We show that while some requirements are achievable methodological standards, others face structural barriers that better methods alone cannot resolve.

**1. Construct definition.** The target construct must be specified independently of the system under test. For welfare, this requires a definition of the latent state $W$ that does not reference model outputs or architecture. No current proposal meets this standard; definitions typically enumerate behavioral or computational indicators that are themselves products of training. *Status: Achievable in principle; unmet in practice.*

**2. Discriminant validity.** The benchmark must distinguish welfare-relevant variation from welfare-irrelevant variation (e.g., prompt format, output language, decoding temperature). Multi-trait multi-method designs could in principle test this, but require that at least some measurement channels are independent of the training pipeline. For biological subjects, physiological measures provide such channels; for AI systems, all channels are downstream of $\theta$. *Status: Achievable in principle; structurally difficult.*

**3. Resistance to optimization.** Retro-holdout testing (Haimes et al., 2024) can quantify whether a benchmark resists gaming when held out from training. This addresses co-engineering (§3) but not construct validity (§4): a held-out benchmark that a model has never seen still lacks downstream consequences that could expose validity failure. *Status: Achievable; addresses §3 but not §4.*

**4. Predictive validity.** A useful benchmark should predict something observable beyond its own scores. Safety benchmarks predict deployment incidents; fairness benchmarks predict disparate impact in hiring or lending. A welfare benchmark would need to predict some real-world outcome tied to the system's well-being. Because the design process and the measurement space share a causally entangled history, no such prediction can distinguish welfare-mediated from design-mediated explanations: any observed outcome is equally consistent with both $H_W$ and $H_D$. There is nothing for the benchmark to be *differentially* checked against. *Status: Structurally unachievable.*

**5. Falsification criterion.** The benchmark must specify what downstream observation would count as evidence that it fails to measure anything genuine. This is our central requirement from §4. Because $H_W$ and $H_D$ are observationally equivalent, no empirical outcome can falsify one relative to the other. Any purported falsification criterion would be equally consistent with both hypotheses and therefore lacks discriminative power. *Status: Structurally unachievable.*

Requirements 1–3 are achievable methodological standards that any serious benchmark effort should adopt. Requirements 4–5, however, are not gaps awaiting better methods. They are consequences of causal unidentifiability (§4): when the design process and the measurement space share a causally entangled history, no benchmark score can serve as a prediction that distinguishes welfare-mediated from design-mediated explanations, and no downstream observation can falsify one hypothesis relative to the other. The benchmark checklist is therefore not a roadmap for incremental progress but a demonstration that welfare benchmarks cannot, in principle, function as institutional gates for systems of this kind (§6.1).

