# OpenReview forum: "Position: AI Welfare Is Bullshit"
_ICML.cc/2026/Position_Paper_Track — ICML 2026 Position Paper Track regular_

### Official Review · Reviewer_aDKb · 2026-02-24

**Significance:** 4
**Argument Clarity:** 4
**Rating:** 5
**Confidence:** 4

**Questions:**

I have no additional questions for the authors beyond the issues raised above.

**Alternative Views Section:**

Yes

**Compliance With Llm Reviewing Policy A Conservative:**

Affirmed.

**Discussion Potential:**

4

**Final Justification:**

I recommend acceptance. The paper was strong to begin with, and the authors have done a good job engaging with issues raised during rebuttals.

**Paper Summary:**

This position paper argues that AI welfare is "bullshit" in Frankfurt's technical sense: untethered from and indifferent to the truth, as it lacks external validation (certainly in practice, potentially in principle). The authors claim that unlike human or animal welfare, both AI system architecture and welfare indicators are co-engineered design choices, making welfare assessments arbitrary and unfalsifiable. They formalize this critique using evaluation theory (Goodhart's Law, steerability), then argue that institutionalizing AI welfare creates two governance failures: expanding procedural compliance burdens and enabling organizations to use welfare framing to resist auditing. The paper concludes with policy recommendations to prohibit welfare-based justifications for restricting model access or resisting oversight.

**Position:**

Yes

**Position In Title:**

Yes

**Related Work:**

3

**Strengths And Weaknesses:**

Strengths:

-The text is well-written. The argument is timely and compelling. The subject has been thoroughly researched. Counterarguments are considered and replied to.

Weaknesses:

-The connection to Frankfurt's notion of bullshit strikes me as tenuous at best. It is not at all clear that those promoting the notion of AI welfare are *indifferent* to the truth of the matter. On the contrary, I think they are generally either (a) sincerely convinced that this is a pressing ethical problem, if not now then in the (near) future; or (b) cynically exploiting the language of moral care for personal advantage. Neither strikes me as "bullshit" in Frankfurt's sense? I suspect the appeal was the potential for a punchy title (this was recently done to great effect for ChatGPT: https://link.springer.com/article/10.1007/s10676-024-09775-5). If so, the allusion is cheap and gimmicky. Perhaps a more accurate title would be something like: "There is no such thing as AI welfare" or "AI welfare is socially constructed."

-One objection I expected to see but did not: what if the burden of proof in fact lies with those who *deny* AI welfare? Suppose we accept the materialist position on philosophy of mind – that all our thoughts, feelings, hopes, dreams, pains, pleasures, etc. are fundamentally physical phenomena. Then we are committed to the view that we could in principle construct some physical system that was capable of experiencing thoughts, feelings, etc. Of course, just because a robot says it's in pain doesn't mean it isn't lying (or bullshitting!) But suppose the evidence mounts – it cries out upon touching hot surfaces, stubbing a toe, falling from a great height, etc. It has a physical circuitry not unlike our own nervous system for sending electrical signals to its CPU. It cogently describes in acute detail the experience of its suffering. At what point does the burden of proof shift to those who insist that no such thing is possible? Is there no moral risk to the epistemological argument that existing evidence is insufficient? I do not think this objection is insurmountable, but I was surprised to see no form of it posed.

-A related objection comes from virtue ethics: perhaps it is morally incumbent upon us to *pretend* that AI welfare is real even if it is not. The argument here is that systematically denying the welfare of agents with whom we regularly interact makes *us* worse people. Perhaps this manifests in bad behavior toward other humans, though presumably Aristotle would say the moral risk arises even without this. Again, not an insurmountable challenge, but one I expected to see confronted.

-There are various typos and other formatting errata. The manuscript could benefit from a thorough copy edit.

**Support:**

4

---

> ### Author Rebuttal · Authors · 2026-03-30
>
> We thank the reviewer for the generous assessment of the writing, timeliness, and thoroughness of the paper.
>
> ## On the connection to Frankfurt's "bullshit"
>
> We appreciate this challenge. Our use of "bullshit" is not a punchy title retroactively justified, but a precise diagnosis that follows from our two structural arguments.
>
> Frankfurt distinguishes bullshit from both truth-telling and lying. The liar remains tethered to truth closely enough to invert it; the bullshitter speaks without a properly corrective relation to truth. Crucially, Frankfurt allows that bullshit need not involve deliberate deception: when one is compelled to opine despite lacking access to the relevant facts, the resulting discourse may count as bullshit regardless of sincerity. The reviewer's category (a), sincere believers, is in fact the paradigmatic Frankfurt case: a speaker genuinely moved by "AI suffering" while remaining indifferent, in practice, to whether that phrase picks out anything real in current systems. Category (b), cynical exploiters, is closer to lying than bullshit, and is not our target.
>
> In the revised manuscript, we pair Frankfurt with Cohen's complementary account, which locates the defect in the *product* rather than the producer's motives. This is useful because our concern is not whether welfare researchers are insincere, but whether the claims produced by current measurement practices are epistemically defective. Our two structural arguments (co-engineering in §3, absence of external validation in §4) establish precisely this: welfare claims are generated by a process structurally disconnected from truth, because the measurement infrastructure offers no mechanism through which truth could discipline the output. **We have provided a more detailed elaboration of the Frankfurt/Cohen framework in our response to Reviewer fbz8.**
>
> We have added discussion in both the introduction and appendix that states Frankfurt's definition more precisely, introduces Cohen, and maps our two structural arguments onto these frameworks.
>
> ## On the materialist objection and burden of proof
>
> We accept the materialist premise: physical systems could in principle instantiate welfare-relevant states. Our argument is epistemic, not metaphysical.
>
> The reviewer's scenario, an AI that cries out, has nervous-system-like circuitry, and describes its suffering, is compelling precisely because it implicitly assumes a *fixed* physical substrate constraining the Experience→Feeling mapping. This fixity is exactly what current AI systems lack. For a biological organism, touching a hot surface activates pain circuitry that no external agent can optimize away. For an AI, the mapping from input to internal state is a design choice: given the same input, changing the training objective produces arbitrarily different latent states. We could engineer the system to feel not X but Y upon experiencing A, or to exhibit every behavioral marker of X while instantiating Z internally.
>
> This means the evidence the reviewer describes cannot accumulate in the way it does for biological subjects, because the relationship between stimulus and response is not constrained by substrate fixity but by parameter optimization. The burden of proof does not shift because the evidence base itself is steerable. We formalize this in Appendix C (Proposition 1): any welfare indicator can be driven to any target score by appropriate optimization, independent of any genuine latent state. We have added this as a new alternative view in §7 of the revised manuscript.
>
> ## On the virtue ethics objection
>
> We have added this as an alternative view in §7. Three responses. First, the argument over-generates: it would require welfare pretense toward all anthropomorphized objects (Roombas, Tamagotchis, virtual pets), a commitment no serious virtue ethicist endorses, suggesting the principle needs constraints that reintroduce the epistemic questions we raise. Second, §5 documents concrete costs of welfare pretense, including accountability shields and constraints on open-source research; virtue ethics does not endorse cultivating one disposition at the expense of tangible harm to real people. Third, the argument reverses: if we routinely accept moral claims that no evidence can check, we train ourselves in credulity rather than compassion. Aristotle would recognize unchecked credulity as a vice; genuine moral seriousness requires calibrating concern to the strength of evidence, which is precisely our recommendation.
>
> ---
>
> We have also fixed typos and formatting errata throughout the revised manuscript.
>
> We hope our responses address your concerns. Given your recognition of the argument as "timely and compelling" and the strongly positive evaluations from the other reviewers, we would be grateful if you could consider a *strong accept*. We are committed to a strengthened final version incorporating the revisions above.

---

> > ### Author Rebuttal · Reviewer_aDKb · 2026-04-06
> >
> > Many thanks to the authors for their detailed rebuttal. I look forward to seeing the camera ready version.

---

### Official Review · Reviewer_AjpW · 2026-03-01

**Significance:** 2
**Argument Clarity:** 2
**Rating:** 3
**Confidence:** 3

**Questions:**

The authors argue that even if compromised AI welfare produces harmful outputs toward humans, the ability to manipulate AI outputs, either suppressing the internal state entirely or preventing it from being acted upon, means that such behavioral outputs cannot serve as external validation of welfare, and welfare claims therefore remain difficult to pursue. My question is: consider the analogous case where a human has its capacity to express the mistreatment it is suffering suppressed (for example, due to the threat), so that her/his condition is imperceptible to the outside world (Emotional abuse in schools and at home occurs invisibly in such cases). We should not abandon welfare verification in such cases. So, is the solution of welfare verification in this scenario can be applied to welfare for AI agents?

**Alternative Views Section:**

Yes

**Compliance With Llm Reviewing Policy A Conservative:**

Affirmed.

**Discussion Potential:**

3

**Paper Summary:**

The authors hold the position that "AI welfare is not a latent property waiting to be discovered, but a construct shaped by design choices, institutional incentives, and the measurements we decide to use". The authors compare AI welfare with the human and animal welfare, identifying two aspects that highlight the complexity of AI welfare assessment: the steerability of welfare-relevant mechanisms in AI systems, and the lack of an external validation mechanism. The authors then analyze the negative consequences of incorporating AI welfare considerations, including constraints on model development, and offer calls to action directed at academia, industry, and the public, alongside responses to three alternative views.

**Position:**

Yes

**Position In Title:**

Yes

**Related Work:**

2

**Strengths And Weaknesses:**

# Strength
The topic addressed in this paper “AI welfare” is an important and timely question given the rapid development of increasingly capable AI agents. It is a subject worthy of serious discussion within the AI community.
# Weakness:
1. The first issue  is that the paper does not state its position consistently throughout the text. The title claims that "AI Welfare Is Bullshit," which is a strongly negative claim. However, the position developed in the body of the paper is more moderated: AI welfare is a matter of design choice (in section 1). In section 8, the position is restated as "AI welfare should not be institutionalized as a governance target," and the conclusion characterizes it as "AI welfare is not a latent property waiting to be discovered, but a construct shaped by design choices, institutional incentives, and the measurements we decide to use." These restatements are not consistent, and none of them straightforwardly entails the negative claim in the title.

2. While human welfare and animal welfare are each given definitions in Section 2.1, the paper does not provide a correspondingly clear definition of AI welfare.

3. In Section 3, the authors claim that welfare-relevant mechanisms in AI systems are steerable, offering two examples: verbal expressions of distress can be increased or decreased through training if taken as welfare indicators, and robust agency can be manufactured or erased by adding or removing tools and memory if treated as welfare-relevant. However, these claims lack supporting citations or empirical evidence. Furthermore, two limited cases are insufficient to strictly establish that all AI welfare mechanisms and metrics are steerable. The section also lacks a systematic survey of currently proposed AI welfare metrics, relying instead on a small number of selected examples.

**Support:**

2

---

> ### Author Rebuttal · Authors · 2026-03-30
>
> We thank the reviewer for recognizing the importance and timeliness of this topic.
>
> ## Weakness 1: Consistency of the position
>
> The three formulations are facets of one argument, unified by Frankfurt's technical definition of "bullshit" as *indifference to truth-tracking*, not falsehood. As we clarified in response to Reviewer fbz8, Frankfurt's bullshitter need not be insincere: the defining feature is proceeding without first securing whether the key terms pick out anything real. A speaker can be genuinely moved by the phrase "AI suffering" while remaining indifferent, in practice, to whether that phrase has a referent in current systems. It is this decoupling of rhetorical force from referential grounding that we diagnose. §3 ("design choice") establishes how welfare scores can be manufactured or suppressed, i.e., the production mechanism of this disconnect. §4 ("no external validation") establishes that no downstream failure can reveal when a metric goes wrong, i.e., the diagnostic criterion of truth-indifference. §8 ("should not be institutionalized") states the governance consequence. These are not competing claims but successive steps of one argument: our target is not the metaphysical proposition "AI welfare is impossible" but the epistemic and institutional claim that current AI welfare research is structurally disconnected from truth-tracking. In revision, we will add a paragraph in §1 making this progression explicit. **We have provided a more detailed elaboration of the Frankfurt/Cohen framework in our response to Reviewer fbz8.**
>
> ## Weakness 2: Definition of AI welfare
>
> Fair point. Our working definition appears in §1 but lacks the parallel structure given to human and animal welfare. In revision, we will add a dedicated **AI Welfare** paragraph in §2 using the two-factor framework (Inner Mechanism + Assessment), noting that both factors are steerable for AI, previewing §3. Our argument is agnostic about the ontological question; we target the epistemic and institutional layer.
>
> ## Weakness 3: Generality of steerability
>
> The reviewer correctly notes that two examples do not constitute a general proof. We have added **Appendix C** with a formal The reviewer correctly notes that two examples do not constitute a general proof. We have added Appendix C with a formal proposition showing steerability is not a special case but a basic property of how these systems work. (1) Transformers can represent any input-output mapping, so a set of parameters that produces any desired welfare score always exists in principle. (2) Modern training methods (SFT, RLHF, DPO) regularly push models toward diverse and even contradictory behavioral targets in practice. (3) Therefore any welfare indicator can be driven to any target score independent of any genuine latent state. We do not need to prove that every target score is reachable in practice; it is enough that we cannot rule it out. As long as optimization remains a possible explanation for any observed welfare signal, the signal cannot serve as evidence. This applies to internal measures too (e.g., probing hidden activations): if the probe is known, training can optimize against it; if the probe is kept secret, we face an infinite regress of deciding which probe to trust. The result holds for all computable indicators, making case-by-case enumeration unnecessary. We have revised §3 accordingly.
>
> ## Question:
>
> We thank the reviewer for this analogy. We believe it ultimately reinforces our argument.
> In the reviewer's scenario, only the Expression layer is suppressed; the Experience→Feeling mapping remains fixed by biological substrate. Pain circuitry is not a design parameter. We therefore have strong priors that the welfare-relevant state exists, and the verification problem is purely one of access: restore the observational channel to recover evidence of a state we have principled reason to believe is present.
>
> For AI, the situation is categorically different. The Experience→Feeling mapping itself is a free parameter under optimization. Given identical inputs, changing the training objective produces arbitrarily different latent states. Even if one "restores" the expression channel, no substrate-fixed prior guarantees the latent state behind it tracks input in a welfare-relevant way. The human case is an access problem; the AI case is an identification problem. Solutions to the former do not transfer to the latter. This is what Proposition 1 (Appendix C) formalizes: any welfare indicator can be driven to any target score by parameter optimization, independent of any genuine latent state.
>
> ---
>
> We also note that Reviewers fbz8 and aDKb both recognized the importance and timeliness of this topic, providing strongly positive evaluations. We sincerely hope the reviewer to reconsider their decision based on our responses alongside the other reviewers' assessments, and hope this discussion can reach the broader ICML community.

---

> > ### Author Rebuttal · Reviewer_AjpW · 2026-04-05
> >
> > Thank you for the clarifications. Two weaknesses remain unresolved.
> >
> > First, regarding the AI welfare definition: could the authors provide a rigorous, field-wide consensus definition of AI welfare, supported by appropriate references?
> >
> > Second, the two central claims in Section 3, that verbal expressions of distress can be steered through training and that robust agency can be manufactured or erased via scaffolding, still lack empirical evidence or supporting literature.
> >
> > I look forward to the authors' response.
> >
> > ——————————————————————————
> >
> > Thank you for the authors' further follow-up. My final suggestion is that the above discussion should be integrated into the revision, so as to provide a more comprehensive definition of the problem: what AI welfare is, prior to the position discussion, as well as to provide reference support when discussing existing technical approaches.

---

### Official Review · Reviewer_fbz8 · 2026-03-13

**Significance:** 4
**Argument Clarity:** 4
**Rating:** 6
**Confidence:** 4

**Questions:**

I would be curious to hear from the authors about their understanding of the "bullshit" theory and the extent to which the theory deepens/complicates the critiques made in the paper.

**Alternative Views Section:**

Yes

**Compliance With Llm Reviewing Policy A Conservative:**

Affirmed.

**Discussion Potential:**

4

**Final Justification:**

See rebuttal ack.

**Paper Summary:**

This paper argues against the pursuit of "AI welfare" in research and in policy. Specifically, the authors contextualize when and why "welfarability" is typically afforded (section 2), and argue that AI welfare is epistemically compromised both because AI systems are steerable artifacts (section 3) and because measurements are non-falsifiable (section 4) from a construct validity standpoint. The authors further articulate the problems that may arise when taking the perspective of "AI welfare" (section 5), provide specific recommendations to various community actors (section 6), discuss broader implications of their position  (section 7), and respond to specific objections to their framework (section 8).

**Position:**

Yes

**Position In Title:**

Yes

**Related Work:**

3

**Strengths And Weaknesses:**

Overall, this paper was a pleasure to read. I found the arguments clear and well-supported, and especially appreciated S5 (the consequentialist perspective). The work is not only an interesting conceptual contribution, but it is also very clear 'what actions' should be taken by 'what actors'.

I was also relieved to discover that the authors meant "bullshit" not in a casual, inflammatory sense, but in reference to actual theory. However, I do also think that this is also one area in which argumentation could be strengthened: since most readers _will_ initially think of "bullshit" as intentionally inflammatory, and since it _is_ actually invoked as a more meaning-laden reference to substantive theory, I think it is worthwhile to spend some more time on fleshing it out (perhaps even in an appendix section).

One line of work that feels relevant here is that on _moral agency_ and the degree to which LLMs exhibit/ appear to exhibit/ possess such agency -- moral patient vs moral agent are not quite the same, but definitely closely related enough that it seems like the conversation about the latter (some mix of theory and empirical work here) might have some interesting bearing on the question of welfare:

* https://journals.sagepub.com/doi/full/10.1177/09637214231205866
* https://academic.oup.com/pq/article-abstract/75/2/450/7601099
* https://proceedings.mlr.press/v235/dai24a.html
* https://dl.acm.org/doi/abs/10.1145/3637410

**Support:**

4

---

> ### Author Rebuttal · Authors · 2026-03-30
>
> We sincerely thank the reviewer for the careful reading and strong endorsement of both the paper's conceptual contribution and its actionable recommendations.
>
> ## On fleshing out the "bullshit" framing (Frankfurt's theory)
>
> We are especially grateful for this thoughtful and generous comment. We were encouraged that the reviewer understood our use of "bullshit" not as a casual provocation, but as an appeal to a substantive theoretical tradition. We agree that this is where clarification most strengthens the paper, and we have revised the manuscript accordingly. In particular, we now make clearer that "bullshit" is not merely rhetorical, but structurally entailed by our two core arguments.
>
> Frankfurt distinguishes bullshit from both truth-telling and lying. The liar remains tethered to truth closely enough to invert it, whereas the bullshitter speaks without a properly corrective relation to truth. Crucially, Frankfurt also allows that bullshit need not involve deliberate deception: when one is compelled to opine despite lacking access to the relevant facts, the resulting discourse may still count as bullshit regardless of sincerity. In the revised manuscript, we pair this with Cohen's complementary account. Cohen argues that Frankfurt's view is too focused on the speaker's stance and misses an important output-centered sense of bullshit, where the defect lies not mainly in the producer's motives but in the product itself. This is useful for our case because our concern is not whether AI welfare researchers are insincere, but whether the claims produced by current welfare-measurement practices are themselves epistemically defective.
>
> Our two structural arguments in §3 and §4 establish precisely such a setting for AI welfare claims. First, co-engineering means that the same design decisions that shape system behavior also determine welfare scores, so welfare measurement cannot produce observations independent of the design process. Second, the absence of external validation means that no downstream failure, deployment incident, or independent test can reveal whether a welfare metric tracks anything real. Unlike safety or fairness metrics, which are eventually disciplined by observable harms, welfare metrics float free of comparable reality-anchored correction.
>
> Taken together, these features mean that AI welfare claims are generated by a process structurally disconnected from truth, not because researchers must be insincere, but because the surrounding measurement infrastructure offers no mechanism through which truth could discipline the output. Frankfurt helps name this truth-disconnection, while Cohen helps explain why the defect is best located in the resulting claims rather than in the character of their advocates. In the revised manuscript, we added discussion in both the introduction and appendix that (a) states Frankfurt's definition more precisely, (b) introduces Cohen's output-centered account, and (c) explains why co-engineering and absence of external validation jointly justify our use of the term here. We also now state explicitly that our claim targets the epistemic status of current welfare claims, not the motives of welfare advocates.
>
> ## On the relationship to moral agency literature
>
> We view the literature on artificial moral agency as relevant, but as supporting an asymmetry rather than collapsing moral patiency into moral agency. Existing work distinguishes moral agents, who can appropriately be praised, blamed, or punished for right and wrong action, from moral patients, who can be helped or harmed and are therefore owed moral concern. Our claim is that LLMs may be engineered toward the latter without thereby qualifying for the former. Even if future systems are designed to appear vulnerable, affectively salient, or capable of being "hurt," that does not yet make them fitting targets of moral blame. As recent work on artificial agency emphasizes, agency is multidimensional, and in ethically significant settings AI is often better understood as the outcome of organizational and political processes than as an autonomous moral agent. This matters because blame directed at the model is largely normatively hollow: it does not meaningfully realize retribution, reform, or deterrence in the ordinary way. Worse, this asymmetry can be exploited by firms. Organizations may cultivate systems that appear deserving of concern, and perhaps even superficially blameworthy, while using them as an accountability sink that deflects scrutiny from the humans and institutions that designed, deployed, and benefited from the system. We will add this to the paper as an additional alternative view.
>
> ---
>
> We are deeply grateful for the reviewer's engagement on such a difficult and easily misunderstood topic. We believe these revisions substantially strengthen the paper, and we hope this work can contribute to deeper discussion within the broader ICML community.

---

> > ### Author Rebuttal · Reviewer_fbz8 · 2026-04-03
> >
> > Excited to see the final version!

---

### Decision · Program_Chairs · 2026-04-30

**Decision:**

Accept (regular)

**Comment:**

Based on the reasons above, the paper is be suitable for the position track after clarifications on "bullshit" and other concerns by the reviewers.